

# Invited Perspectives: Integrating hydrologic information into the next generation of landslide early warning systems

Benjamin B. Mirus[1,2], Thom Bogaard[3], Roberto Greco[4], Manfred Stähli[2]

[1] United States Geological Survey (USGS), Geologic Hazards Science Center, Golden, Colorado, USA
[2] Swiss Federal Institute for Forest, Snow, and Landscape Research (WSL), Birmensdorf, Switzerland
[3] Department Water Management, Delft University of Technology, Delft, The Netherlands
[4] Department of Engineering, University of Campania "Luigi Vanvitelli", Aversa, Italy

*Correspondence to*: Ben Mirus (bbmirus@usgs.gov)

**Abstract.** Although rainfall-triggered landslides are initiated by subsurface hydro-mechanical processes related to the loading, weakening, and eventual failure of slope materials, most landslide early warning systems (LEWS) have relied solely on rainfall event information. In previous decades, several studies demonstrated the value of integrating proxies for subsurface hydrologic information to improve rainfall-based forecasting of shallow landslides. More recently, broader access to commercial sensors and telemetry for real-time data transmission has invigorated new research into hydrometeorological
thresholds for LEWS. Given the increasing number of studies across the globe using hydrologic monitoring, mathematical modeling, or both in combination, it is now possible to make some insights into the advantages versus limitations of this approach. The extensive progress demonstrates the value of in situ hydrologic information for reducing both failed and false alarms, through the ability to characterize infiltration during, as well as the drainage and drying processes between major storm events. There are also some areas for caution surrounding the long-term sustainability of subsurface monitoring in
landslide-prone terrain, as well as unresolved questions in hillslope hydrologic modeling, which relies heavily on the assumptions of diffuse flow and vertical infiltration but often ignores preferential flow and lateral drainage. Here, we share a collective perspective based on our previous collaborative work across Europe, North America, Africa, and Asia to discuss these challenges and provide some guidelines for integrating knowledge of hydrology and climate into the next generation of LEWS. We propose that the greatest opportunity for improvement is through a measure-and-model approach to develop an
understanding of landslide hydro-climatology that accounts for local controls on subsurface storage dynamics. Additionally, new efforts focused on the subsurface hydrology are complementary to existing rainfall-based methods, so leveraging these with near-term precipitation forecasts is also a priority for increasing lead times.

## 1 Subsurface hydrologic information improves landslide forecasting

Hydrology plays an important role in shallow landslide initiation (Campbell, 1975; Lu and Godt, 2013; Bogaard and Greco,
2016); this has been demonstrated through many decades of monitoring hydrologic response and slope stability on individual hillslopes and zero-order basins around the world (e.g., Sidle and Swanston, 1982; Sidle and Tsuboyama, 1992; Torres et al., 1998; Godt et al., 2009; De Vita et al., 2013; Liang, 2020; Marino et al., 2020; Ashland, 2021). These observations are supported by well-established theory in soil physics and geomechanics, whereby the addition of water to porous media changes their strength and weight, contributing to a force imbalance and triggering slope failure (e.g., Terzaghi, 1943).
Theoretical advances have supported the development of mathematical models to numerically simulate – with varying degrees of complexity – the conditions leading up to these critical conditions (e.g., Montgomery and Dietrich, 1994; Terlien, 1997; Van Beek, 2002; Brien and Reid, 2008; Baum et al., 2010; Lehmann and Or, 2012). Despite this conceptual understanding and advanced model development, most local and regional landslide early warning systems (LEWS) rely on rainfall inputs alone, typically with the well-worn intensity-duration (ID) threshold approach (Caine, 1980; Guzzetti et al.,
2008; Brunetti et al., 2018; Segoni et al., 2018) and the related event-duration (ED) threshold (Innes, 1983; Guzzetti et al., 2020). These are built upon the assumption that if it rains hard enough for long enough in a landslide-prone area, the storm



event will trigger slope failures. When event-based rainfall thresholds are used alone, the hydrologic conditions preceding the triggering and the associated antecedent wetness have no bearing on predicted slope stability. The negligible role of previous rainfall, evapotranspiration, and hillslope drainage may or may not be true based on local variations (e.g., Thomas

et al., 2020). Furthermore, these underlying assumptions must be questioned in the context of a changing climate and non-static (a)biotic terrain conditions (Ehret et al, 2014), where multiple competing factors related to infiltration, drainage, and evapotranspiration interact to influence predisposing factors and triggering conditions (Gariano and Guzzetti, 2016; Jakob, 2022).

Although broadly applicable, with many centuries of rainfall data underpinning its implementation, the generalized ID approaches rely on several conceptual flaws (refer to Bogaard and Greco, 2018) and lack specificity. As the benchmark standard, these approaches have succumbed to inertia with few novel methodological advances since their early inception (e.g., Caine, 1980). Still, a handful of studies over many decades and across a variety of settings have shown that using rainfall data to develop well-informed proxies for seasonality or hillslope antecedent wetness can improve landslide

prediction with ID thresholds (e.g., Campbell, 1975; Wilson and Wiezorek, 1995; Crozier, 1999; Glade, 2000; Godt et al., 2006; Napolitano et al., 2016). Calculation of these proxies often reflect the basics of infiltration and soil-water storage, but consistently fall short of capturing the complex wetting and drainage dynamics observed in the variably saturated near surface. Similarly, satellite and remote sensing products capture the seasonal shifts in landscape wetness that are broadly relevant for landslide potential (Felsberg et al., 2021; Zhao et al., 2021; Distefano et al., 2023), but their coarse resolution

and considerable latency fail to capture the rapid subsurface dynamics on hillslopes that are critical to forecasting landslide potential (Thomas et al., 2019). Electrical resistivity tomography has revealed nuances related to subsurface moisture patterns in landslide settings (e.g., Perrone et al., 2014; Uhlemann et al. 2017), but these hydrogeophysical methods remain cumbersome to implement and their sampling rates are currently too slow to capture the rapid unsaturated zone responses that trigger shallow landslides (Nimmo et al., 2021). Recent progress with automated empirical modeling shows some

promise in recreating hillslope hydrologic response (Orland et al., 2020) and highlight the importance of rainfall over specific terrain attributes in predicting spatiotemporal populations of landslides (Mondini et al., 2023). However, in an uncertain future with increasing landscape disturbances, climate change, and non-stationary responses in hydrologic systems, the next generation of LEWS can be advanced through incorporating our mechanistic understanding the hydroclimatology of triggering conditions.


Recently, the emergence of the "Internet of Things" has provided further motivation for integrating hydrologic information to improving LEWS predictive performance because subsurface monitoring data can be accessed in real-time to understand evolving hillslope wetness conditions (Mirus et al., 2018a; Abraham et al., 2020; Piciullo et al., 2022). This approach has the potential to outperform rainfall-based estimates of antecedent wetness and imminent triggering conditions because in situ

data can capture the true hillslope hydrologic response associated with landslide initiation. A handful of studies that integrate different types of subsurface measurements directly into landslide initiation thresholds show some promising results (Mirus et al., 2018b; Zhao et al., 2019; Marino et al., 2020; Wicki et al., 2020; Abraham et al. 2021; Pecoraro and Calvello, 2021); their success reflects the understanding of the relevant hydrologic processes for their region of interest. Despite the many advances and limitations of current approaches to LEWS (refer to reviews by Guzzetti et al., 2012, 2020; Stähli et al., 2015;

Piciullo et al., 2018), new research on real-time hydrometeorological thresholds is still an emerging field (Greco et al., 2023). No guidelines have been established for developing a reliable LEWS that is informed, at least in part, by real-time hydrological information. Considering the first action in the Kyoto Landslide Commitment 2020 involves improving the precision and reliability of landslide warning (Sassa et al., 2023), we propose that integrating insights from in situ hydrologic measurements into LEWS is essential.




### 1.1 Continuous field monitoring for comprehension of triggering processes

The complex interaction between hydrological and mechanical processes results in many possible ways of adding water to transition from stable to unstable conditions, leading to very different types of triggering conditions for seemingly similar settings (e.g., Fusco et al., 2022). These depend largely on how geology, climate, geomorphology, vegetation, and landscape

disturbances have influenced the geometry and hydromechanical properties of soils, vegetation distribution, and the geometry of hillslope source areas where landslides initiate (Sidle et al., 2017). Every hillslope is unique, so we cannot characterize the true subsurface heterogeneity and corresponding controls on landslide triggering across a landscape, but hydrological information provides a foundation of comprehension to inform landslide forecasting across contrasting locations. For example, in areas with strong seasonality the antecedent soil-moisture conditions may be critical for refining

landslide initiation thresholds (e.g., Godt et al., 2006; Mirus et al., 2018a,b; Thomas et al., 2018a; Wicki et al., 2020, Marino et al., 2021). In contrast, some regions typically remain quite wet, and prior conditions seem to add very limited value in constraining landslide potential (Thomas et al., 2020; Patton et al., 2023). During the most recent decades, soil moisture has become increasingly easy to monitor in situ, but other measured variables such as groundwater levels have also been used quite effectively (Wei et al., 2019; Wei et al., 2020; Illien et al. 2021; Marino et al. 2021; Uwihirwe et al., 2022; Roman

Quintero et al., 2023). As a state variable, analysis of volumetric water content profiles can reveal whether the soil is accommodating infiltration through changes in storage or if it has exceeded field capacity and is allowing more rapid vertical fluxes to the saturated zone below. In contrast, shallow groundwater fluctuations (when they can be measured) reflect not just water added from vertical infiltration, but also the three-dimensional (3D) subsurface flow field from upslope accumulation to downslope drainage. Of course, other hydrologic state variables may be used as proxies for antecedent

conditions such as snowmelt (Mostbauer et al., 2018; Hinds et al., 2019; Wayllace et al., 2019) and catchment storage (Ciavolella et al., 2016; Marino et al., 2022).

It is difficult to know a priori which conditions and variables are important for an area of interest, but even a few years of hydrologic monitoring can improve understanding of the variably saturated hillslope responses in stormflow generation (e.g.,

Beven, 2012; Blume and van Mereveld, 2015) and landslide initiation or reactivation (e.g., Godt et al., 2009; Mirus et al., 2017). Accounting for regionally specific controls on infiltration and hillslope drainage dynamics could help improve hydrometeorological thresholds by reducing failed alarms, such as those produced by relatively modest storms on already very wet soils, as well as lowering the number of false alarms, such as those related to heavy precipitation on dry soils. Therefore, to develop the next generation of LEWS, expanding hillslope hydrologic monitoring from a handful of existing

networks to a wider variety of landslide prone terrain worldwide would be highly beneficial. The potential value of long-term hydrologic measurements can be inferred from recent advances in characterizing landslide triggering based on identifying rainfall anomalies or recurrence intervals (e.g., Kirschbaum and Stanley, 2018; Marc et al., 2022). A greatly expanded and openly accessible network of hydrologic observations would further supplement such approaches using relative hillslope wetness to support new inferences about triggering potential.


However, established guidelines for landslide hydrological monitoring are lacking, and providing general advice on how to select appropriate sites and instrumentation equipment is exceptionally difficult. This challenge is compounded by subsurface heterogeneity that cannot be known a priori and logistical considerations that often influence instrument placement (e.g., site access and safety consideration on steep slopes). These nuances are not emphasized in publications or

presentations, so in practice the many seemingly subjective elements in such studies reflect the crucial role of expert judgement. Thus, a first step for hydrologic monitoring is developing a strong conceptual model of local conditions at the



site of interest, based on available observations including geologic maps, soil classifications, landscape morphology, climate, and even records of rainfall and streamflow from the region. We also stress that measurement of the specific hydrologic state variables (i.e., soil moisture, versus groundwater levels, versus soil suction) or the precise values is not critical. Instead,
measurements that capture the relative change in hillslope wetness conditions often provide the most informative variables (e.g., Wicki et al., 2020). In particular, the most useful measurements reflect the widest variability during and between landslide events, which reveals the most about hillslope storage dynamics.

The influence of spatial variability – as well as the difficulty in deciding what to measure and where – can be demonstrated
anecdotally using data from the USGS landslide monitoring site in Sitka, Alaska (Smith et al., 2023). This includes volumetric soil water content and positive pore-water pressures measured in two shallow soil pits that are less than 10 meters apart from each other on a steep hillslope. The period shown (Figure 1) encompasses responses to several major storm events that triggered landsliding across the region, one of which ultimately culminated in a fatal landslide in Haines, Alaska (refer to Darrow et al., 2022), roughly 200 kilometers to the northeast of the monitoring site in Sitka. In Soil Pit 1, the matric
potential and soil-moisture sensors show some variations in near-surface conditions and flashy piezometer response only during peak rainfall. Less than 10 meters downslope, the near surface in Soil Pit 2 is persistently wet with continuous shallow groundwater fluctuations throughout the period. Thus, for informing LEWS development, the largely absent pore pressure measurements in Soil Pit 1 might seem of limited value compared to the soil moisture record for assessing antecedent conditions. In contrast, Soil Pit 2 clearly shows the gradual elevation in groundwater levels for successive
landslide producing storms, but consistently high soil moisture values with no valuable information. Thus, the value of soil moisture in the unsaturated zone versus pore-water pressures in the shallow saturated zone would depend entirely upon the location of the soil pit within the landscape, which is difficult to assess from the landscape position alone. However, together, instrumentation in these two soil pits reveals a potential mechanistic explanation for the shallow landslides and debris flows around Sitka: well drained hillslopes remain consistently wet and support disconnected zones of perched
saturation, so landslides tend to occur once these perched saturated zones connect across broader areas of steep hillslopes and their drainage capacity is overwhelmed by consistently high rainfall input.

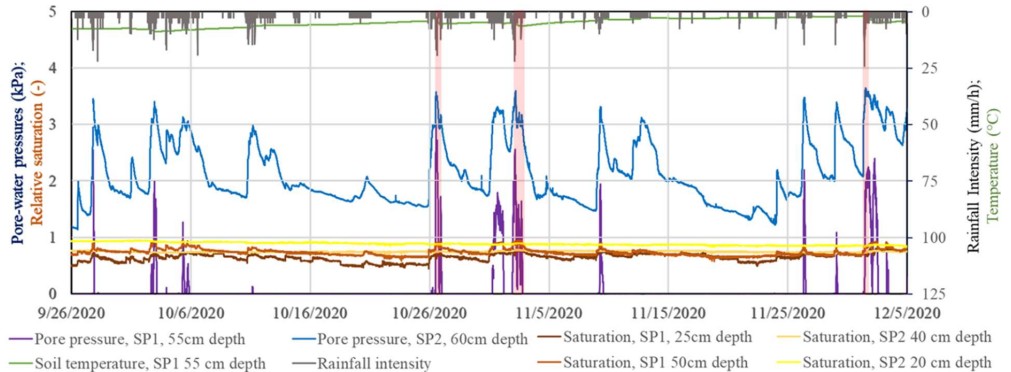

**Figure 1: Climatic conditions and hydrologic variables in two soil pits (SP1 and SP2) on a steep hillslope above Sitka, in southeast Alaska, observed during a sequence of large storm events in October - December 2020; approximate timing of landsliding across the region is shown by the red-transparent bars. Note: near-real-time plots of data are available for situational awareness at https://usgs.gov/programs/landslide-hazards/science/sitka-ak.**



## 2 Understanding the current limits to hydrometeorological thresholds

The major benefits for improving comprehension can be tempered by acknowledging several under-appreciated issues with using in situ monitoring for landslide forecasting. First, the return period of relatively infrequent (but highly destructive) landslide events is often longer than the typical life expectancy of subsurface hydrologic monitoring equipment. Broad estimates of the rainfall recurrence needed to trigger widespread landsliding events across the globe range from a variable 95th percentile (Kirschbaum and Stanley, 2018) to 10-year anomalies (Marc et al., 2022), but on these timescales,

instrumentation often require substantial maintenance that is resource intensive or may exceed their manufacturer-reported life-expectancies, experience the onset of electronic drift, or succumb to destruction from wildlife or vandalism. Indeed, there are substantial elements of chance that allowed previous researchers to capture the hydrologic response conditions during a natural landslide initiation event (Montgomery et al., 2002; Godt et al., 2009; Mirus et al., 2017). Whereas instruments to measure groundwater tables, streamflow, and precipitation can potentially be replaced as needed, volumetric

water content sensors must be placed in relatively undisturbed soil (Caldwell et al., 2022), which means that when sensors fail, they cannot be readily exchanged in the same place without disturbances to the porous media itself. Installation of pore-pressure sensors vary, with some favoring direct contact with the porous media (e.g., Smith et al., 2023) and others using standpipe piezometers (e.g., Thomas et al., 2018b) that potentially allow sensors to be exchanged more easily. Although interoperability across measurement types presents a potential challenge to the long-term sustainability of a global near-real-

time hydrologic monitoring network to inform LEWS, useful hydrometeorological thresholds can be developed by intelligently leveraging existing hydrologic monitoring networks. In particular, observed records can be extended with a measure-and-model approach, whereby shorter periods of data along with measured hydraulic and mechanical properties can inform robust models of the hydrologic conditions related to slope failure (e.g., Ebel et al., 2008; Thomas et al., 2018a; Wicki et al., 2021; Uwihirwe et al., 2022). Even if hillslope hydrologic modeling has its own problem areas (refer to

discussion below) it is likely the most sustainable way to synthesize hydrologic information across space and time.

Second, we accept that in situ subsurface instrumentation favors monitoring precise variations in time rather than capturing broader spatial patterns, and therefore shares similar limitations to any other point measurements, such as rain gages. As with all measurement networks, some expert judgement is needed to design and select a site for subsurface instrumentation,

which influences the data collected and corresponding conclusions that can be inferred. Numerous studies related to subsurface stormflow response used distributed measurements to identify very localized processes that govern hillslope-hydrologic responses such as the role of irregular subsurface topography on "fill-and-spill" processes (Tromp-van Meerveld and McDonnell, 2011), heterogeneous soil profiles and weathering (Zimmer and Gannon, 2018), and preferential flow (Beven and Germann, 1982). However, the same is the case for rainfall measurements, which must rely on radar and satellite

estimates of rainfall variability to determine spatial patterns at relatively coarse scale.

### 2.1 Extrapolating across spatial and temporal scales

It may be even more difficult to extrapolate subsurface hydrologic response dynamics derived from one soil profile across a heterogeneous landscape than atmospheric processes such as rainfall intensities, so some clear advances in remote sensing methods may ultimately be informative for landslide modeling. At the scale of tens to hundreds of meters, cosmic ray

neutron sensors have the capacity to estimate changes in relative hillslope wetness between storms (e.g., Franke et al., 2022), yet these methods remain largely untested for landslide studies in steep, densely vegetated terrain. At the scale of tens to hundreds of kilometers, distributed estimates of soil moisture from remote sensing provides global coverage but falls short of capturing the temporal variations observed in situ that are critical for precise landslide forecasting (e.g., Thomas et al., 2019). Despite the unknowable heterogeneity in the subsurface, one of the greatest opportunities for hydrometeorological threshold



improvement is to use accurate temporal dynamics from in situ observations to inform improved landslide modeling across the larger spatial footprints at hillslope, watershed, and regional scales. A mechanistic understanding of landslide triggering conditions associated with different hydrologic responses would also help constrain the spatial extent over which different hydrometeorological thresholds apply.

**2.2 Alternative hydrometeorological threshold formulations**

Selection of statistical criteria for hydrometeorological threshold optimization can influence the balance between failed and false alarms for a given threshold formulation (e.g., Conrad et al., 2021), but a thorough discussion of uncertainty and performance criteria (e.g., Piciullo et al., 2020) is outside of the scope of this perspective. Instead, we address some informative contrasts between the formats of hydrometeorological thresholds derived using contrasting methods and data inputs (Figure 2). The deterministic threshold (Fig. 2, purple) is based on millions of simulated events from a one-

dimensional (1D) infiltration model calibrated using a few positive pore-water pressure measurements and evaluated with landsliding events from the San Francisco Bay Area, California (Thomas et al., 2018a). This threshold lacks a functional format, but one could theoretically be developed for its convex form, with rapidly decreasing stability at higher antecedent wetness and unconditionally unstable conditions above roughly 0.6 saturation. The format of a bilinear threshold (Fig. 2, blue) was identified empirically and optimized using receiver operating characteristics with numerous landslide events and

years of hydrologic monitoring in the Pacific Northwest of Washington and Oregon (Mirus et al., 2018a,b). It is certainly over-simplified, but the convenient functional format has led to its implementation in other settings such as data sparse Rwanda (Uwihirwe et al. 2022) and testing of satellite-based thresholds in data-rich parts of California (Thomas et al, 2019).

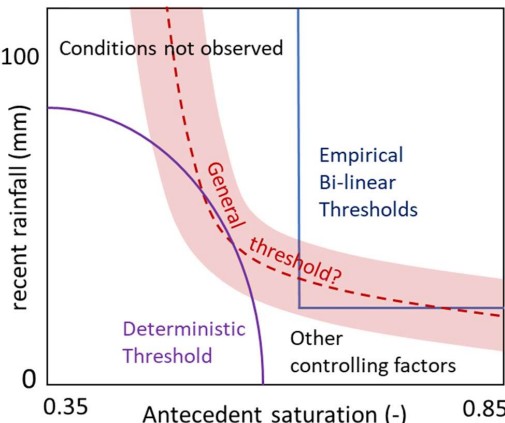

**Figure 2: Different formats of hydrometeorological thresholds for landslide initiation developed based on empirical interpretation (blue) shown with bi-linear thresholds optimized using several years of monitoring data and landslide inventories (Mirus et al., 2018b), versus theoretical understanding (purple) based on a deterministic method using infiltration modeling and millions of synthetic storm events (Thomas et al., 2018a). The deterministic threshold shows unconditionally unstable conditions for excessive rainfall or high saturation levels. In contrast, empirical thresholds indicate that excessive rainfall on dry soils will not trigger**

**failures, whereas even at highest observed saturations moderate rainfall is still needed to trigger failures. Heuristically, a general threshold that accommodates both theoretical understanding and empirical observations seems more reasonable.**

The insight that the deterministic versus empirical approaches support entirely different functional formats of hydrometeorological thresholds raises the question of whether one or the other is more correct, and why they vary so

distinctly. The deterministic thresholds reflect our limited ability to quantify the complex storage dynamics during landslide events, and not necessarily all the processes relevant to infiltration and drainage, which results in considerably broad



conditions that are unconditionally unstable. On the other hand, the empirical threshold format was selected based only on the conditions we have observed thus far and does not necessarily represent all the relevant possibilities. In a recent paper, Palazzolo et al. (2023) propose a hydrometeorological threshold using antecedent soil moisture versus rainfall intensity that

exhibits a similar format to our deterministic threshold at low antecedent wetness combined with a constant rainfall intensity cutoff for higher saturations just like the empirical bilinear threshold. Earlier studies had tested linear models that are similar to the simple formulation used in rainfall-only thresholds, which also led to improvements (Mirus et al., 2018a; Marino et al., 2020).

Overall, these sparse examples of contrasting formats represent just some of the results emerging from recent research into hydrometeorological thresholds, but they raise two important issues. First, that there are observed storage and drainage processes that our hydrological models do not capture (i.e., known conceptual limitations), and second that there are conditions we have not yet observed and are difficult to predict (i.e., unknown range of responses). For example, when the vast majority of both triggering and non-triggering events exhibit correlation between antecedent wetness and rainfall

accumulation (e.g., Scheevel et al. 2017; Palazzolo et al., 2023), it is particularly difficult to project responses to unprecedented rainfall on very dry antecedent soils. Heuristically, we would expect a more generalized threshold that considers the uncertainty in triggering conditions and reflects our conceptual understanding that either more rain or wetter antecedent conditions should each increase the likelihood of landsliding (Fig. 2, red). However, it may be that further studies reveal no universally superior format for hydrometeorological thresholds, and that local practices and priorities will

determine what is used operationally, depending on data availability, system expectations, and risk tolerances.

### 2.3 Limits of process understanding

Three-dimensional, fully coupled surface-subsurface hydrologic models have emerged as a relatively new method for quantifying the hydrogeomorphic processes that contribute to landslide initiation (Loague et al., 2006; Mirus et al., 2007; Ebel et al., 2008). Now, extensive databases on soil geometry and textural classifications (e.g., SSURGO, 2024), novel

pedotransfer functions (e.g., Lehmann et al., 2021), high-resolution continental-scale DEMS (e.g., 3DEP: USGS, 2019), and comprehensive precipitation databases (e.g., IMERG: Huffman et al., 2015) could, in theory, facilitate using physics-based approaches, both conceptual and deterministic, to develop hydrometeorological thresholds for settings all over the globe (e.g., Thomas et al., 2018a; Fusco et al., 2019; Lehmann et al., 2019). However, theoretical gaps in hillslope hydrologic modeling remain a major obstacle. Well-calibrated infiltration models designed to capture observed landslide initiation

processes still struggle to simulate the continuous soil moisture dynamics between events (Thomas et al., 2018c; Wicki et al., 2021; Piciullo et al., 2022) or the influence of complex soil structures (Mirus, 2015; Fatichi et al., 2020).

Regardless of their high computational expense and data demands for parameterization, the reality is we still do not fully comprehend the physics of variably saturated subsurface flow through complex landscapes. The equation used to simulate

diffusive flow through variably saturated soils has been around for quite some time (Richards, 1931), but infiltration models based on the Richards' equation cannot produce results that are consistent with observations of non-sequential wetting fronts (e.g., Graham and Lin, 2011) or preferential flow in the unsaturated zone (Nimmo, 2012, 2020; Beven and Germann, 1982, 2013). These localized hydrological processes combined with subsurface heterogeneity may explain why some hillslopes fail and why other adjacent slopes remain stable, and hence may explain some degree of the variability and uncertainty in

landslide triggering conditions. Such questions remain largely academic as this degree of detail is not necessary (or even achievable) to provide useful and actionable information for landslide loss reduction. Instead, we can focus on more practical modeling and comprehension of hydroclimatology with representative state variables to reduce failed and false alarms.



Ultimately, landslides occur at some point after the initiation zones fill up with infiltration at a rate faster than they can drain.
Although the infiltration component of landslide triggering largely involves largely vertical 1D percolation of precipitation
through unsaturated soils, stormflow responses involve 3D processes including lateral flow diversion and drainage that can
either enhance or reduce landslide potential. These processes are controlled not just within the soils where shallow landslides
initiate, but by complex flow paths in the underlying saprolite, weathered bedrock, and even fracture flow through intact
bedrock. Although robust physically based models still struggle to fully capture those two competing processes due to both a
lack of data and a lack of process understanding, further expansion of simpler conceptual models that are informed entirely
by monitoring data such as a leaky bucket (e.g., Wilson and Wiezorek, 1995), wetness indices (e.g., Godt et al., 2006),
source-responsive methods (e.g., Mirus and Nimmo, 2013) or empirical approaches (e.g., Orland et al. 2020) may be
beneficial. These would have the greatest impact if they can leverage long-term monitoring data to capture the critical
conditions when vertical infiltration exceeds drainage during major storm events, as well as the effect of the landscape-scale
storage dynamics during and between such events. Further iterations of measurement-model comparisons would be helpful
to determine how we can better represent these two competing infiltration and drainage processes in a way that is
representative enough to improve landslide forecasting.

## 2.4 Limits of observational datasets

The wide availability of precipitation records, going back centuries in many cases, paired with the limited frequency of
landslide events, might lead many to believe that there are no surprises and that rainfall thresholds are the most robust and
achievable route to inform LEWS. However, the challenges with balancing failed and false alarms in virtually all landslide
forecasts indicate that there are processes we have not understood and conditions we have not yet observed. For example, the
hydrometeorological threshold models discussed in Figure 2 were developed without any observations of very large storms
on very dry soils, and yet that is potentially the combination of conditions that may influence shallow landsliding in a
changing climate (Gariano and Guzzetti, 2016).

Indeed, the literature includes many different assertions about what to expect in a warming world for different flavors of
landslides (Cannon and DeGraff, 2009; Bennett et al., 2016; Coe et al., 2016, 2018; Parker et al., 2016; Mirus et al., 2017;
Handwerger et al., 2019; Kirschbaum et al., 2020). The degree to which drier soils from elevated evapotranspiration may
balance out the effects of increased frequency and intensity of extreme meteorological events remains unclear. Thus, an
important challenge is to develop mechanistic understanding that would be applicable in a changing climate to inform LEWS
(e.g., Ehret et al., 2014). This includes understanding of how the combination of hydraulic properties such as water-retention
curves, porosity, and saturated hydraulic conductivity influence hillslope storage and drainage dynamics under different
topographic settings (e.g., Mirus and Loague, 2013), or how soil grainsize and mineralogy affect mechanical properties such
as suction stress, cohesion, and internal friction angle (e.g., Lehmann et al., 2021). As more studies use a combination of
measurements and modeling to evaluate the local and regional controls on hillslope hydrologic conditions, it seems likely
that we will identify specific threshold formulations needed for different types of hydroclimatic and environmental settings.
In the same way that the rainfall intensity-duration approach does not apply universally well across the globe (Caine, 1980;
Guzzetti et al., 2008; Baum and Godt, 2010), we could ultimately learn more about the variability and applicability of
hydrometeorological thresholds.



## 3 Towards improved landslide forecasting models

Despite these notable challenges, we maintain that integrating knowledge of hillslope hydrologic processes into landslide forecasting tools is a very promising path forward. Considering that both monitoring systems and models are always imperfect, and that they provide an incomplete picture of the reality, a major challenge is to find a way, with limited understanding and even more limited data availability, to reduce LEWS errors. In some cases, hydrology may not add valuable information over rainfall alone, but in many other settings, we expect that further research can identify hydrologic variables linked to landscape-scale processes that reflect the geologic, geomorphic, and climatic controls on predisposing conditions. Capturing those factors would help improve advance warning for potential landslide conditions prior to short-term forecasts of the precipitation that could ultimately trigger failures locally. With realistic expectations, careful considerations of the issues we outlined above can serve as a framework for a systematic and reliable way to integrate hydrometeorological thresholds into improved local-scale LEWS.

In contrast to accurate hyper-localized LEWS that are currently within reach (e.g., Patton et al. 2023), the transition towards uniform, regional-scale systems with meaningful spatial coverage would require further testing and new methods for interpolating between, and extrapolating beyond, sparse existing observations. In this context, the following efforts provide promising paths towards improving landslide forecasting models worldwide:

1. Determine regional controls on landslide hydroclimatology. This involves assessing the potential infiltration conditions that can influence landslide triggering, including snowmelt (Hinds et al., 2019), prolonged storms (Coe et al., 2015), steady and frequent rainfall (Chleborad et al., 2008) or simply high-intensity bursts of precipitation (Caine, 1980). This also includes quantitatively characterizing landslide seasonality (e.g., Luna and Korup, 2022) and how those seasons may change in the future for different types of slope failure (Jakob, 2022).

2. Develop objective methods to identify the state variables and time scales of interest (e.g., Conrad et al., 2021). In particular, how can we most effectively separate the continuous transition from antecedent versus triggering conditions to improve LEWS performance? Because major obstacles to forecasting subsurface hydrologic conditions remain, it is important to identify which hydrologic factors can be leveraged effectively with the time scales supported by quantitative precipitation forecasts (e.g., Patton et al., 2023).

3. Explore what currently available hydrogeologic information can reveal about subsurface responses. This includes further investigation of important differences between flat versus steep terrain (e.g., Wicki et al., 2020, 2023) and satellite versus in situ hydrologic information (e.g., Thomas et al., 2019) to better leverage existing worldwide monitoring networks. It may be particularly important to explore hydrologic information at a spatial scale that accords with the landslide release extent (rather than relying on single sensors or remotely sensed estimates). This may involve further exploration of emerging technologies (e.g., Franke et al., 2022) or the use of multiple sensors to characterize variably saturated conditions along both vertical and longitudinal hillslope profiles (e.g., Mirus et al., 2017). It is certainly more promising to look at relative changes in hydrologic metrics than to seek absolute threshold values.

4. Establish a global repository of rainfall-triggered landslide inventories with associated hydrologic information. Open access to these data will facilitate research and synthesis with different conceptual modeling approaches across different hydroclimatic and environmental conditions. Then, through a coordinated measure-and-model approach, researchers can test generalized methods to extrapolate controls on infiltration and drainage dynamics across a range of realistic landslide triggering conditions.



Beyond these potential research opportunities, we close with a few practical considerations. There is extensive debate about what "early" means in the context of actionable information for an operational LEWS, which is a discussion reserved for another venue. Instead, it is important to consider that any LEWS that relies only on currently observed conditions, whether precipitation, hydrology, or even detection of incipient slope movement, is largely limited to "now-casting" rather than forecasting. Hydrologic state variables reflect the subsurface response to recent water inputs, and for this reason these

observations should be most accurately indicative of failure (Figure 1). Precipitation precedes infiltration and represents the forcing conditions immediately prior to the hydrologic response when landslides are imminent. To effectively leverage these two sources of information and maximize the "Early" in LEWS, hydrometeorological thresholds ideally would rely on some modeling, whether simple or complex, that starts with recent hydrologic conditions and predicts the effects of forecasted precipitation on slope stability, and potentially assimilates monitoring data to update such forecasts in near-real-time.


At this point we can definitively state that integrating hydrologic information has led to improvements in landslide forecasting over existing LEWS model formats, but are the additional investments in data and research needed for these universally justified for operational systems? Ultimately, it is unclear how well we can expect any LEWS to perform barring other scientific advances. For example, we are not aware of any operational or research-oriented landslide forecasting

approach that successfully accounts for the spatially variable rainfall, triggering conditions, or the inherent uncertainties in short-range quantitative precipitation forecasts. Although LEWS may remain an imperfect tool due to the inherently stochastic and episodic nature of landslide initiation, advancing our understanding of hillslope hydrology across different climatological and geologic settings is well within reach and could soon lead to improved landslide forecasting models globally.

**Author Contributions**

BBM, TB, RG, and MS contributed to the discussion and conceptualization of the perspectives herein. BBM performed the data curation and visualization. BBM prepared the original manuscript with contributions and editing from all co-authors.

**Competing Interests**

The authors declare that they have no conflict of interest.

**Acknowledgements**

The authors would like to thank Dennis Staley and Brian Collins for their constructive reviews of a previous version of this manuscript. The first author is grateful for the WSL Visiting Fellowship that helped facilitate this perspectives paper.

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



Early warning of increased landslide potential provides situational awareness to reduce landslide-related losses from major storm events. For decades, landslide forecasts relied on rainfall data alone, but recent research points to the value of hydrologic information for improving predictions. In this article, we provide our perspectives on the value and limitations of

690 integrating subsurface hillslope hydrologic monitoring data and mathematical modeling for more accurate landslide forecasts.