# Peer review of "Invited Perspectives: Integrating hydrologic information into the next generation of landslide early warning systems"

_EGUsphere, 2024_

## Author Comment (AC1)

Dear Luigi,

Thank you very much for the candid and overall favorable review. We are very pleased that you enjoyed reading our perspective and appreciate your valid criticisms. We are grateful to have this stimulating open discussion with such an esteemed expert in the field.

I (Ben) fully accept your concern that I have been overly autoreferential and it is a fair point that there are several sentences where alternative (or additional) citations could be appropriate. However, I should note that the vast majority of my self-citations (15!) are led by students, postdocs, and other coauthors, whose work I am eager to draw more attention to. None-the-less, it is possible I have over-interpreted the "perspective" element of this piece, and of course I am most familiar with papers my coauthors and I have published. If this leaves you with an unpleasant impression, surely at least a few others will agree. To avoid this, I followed your suggestion and carefully read through to determine when I could work harder to find alternative citations, versus when a self-citation is indeed necessary or the most appropriate option. For example, we can cite alternative papers for the source-responsive model (i.e., Nimmo and Mitchell, *VZJ*, 2013 instead of Mirus and Nimmo, *WRR*, 2013); in contrast, there is no replacement for the Sitka dataset (Smith et al., *USGS,* 2023). As you noted, some self-citations "are justifiable (if not due)" for this type of perspectives piece, but I am confident that I could cut the number down considerably, especially limiting my own first-authored citations to just a few. Additionally, I will be sure that whenever possible, relevant self-citations are accompanied by any other complimentary works (i.e., in addition to Mirus and Loague, *WRR*, 2013, add Lanni et al., *ESPL*, 2013; or in addition to Lehmann et al., *GRL,* 2021, add Van Looy et al., *Rev. Geophys.*, 2017).

Regarding your other very constructive suggestion of adding more discussion on data-driven approaches, my coauthors and I agree that many of these approaches have considerable merit for process-based landslide forecasting (e.g., one self-reference to Orland et al., *GRL*, 2020). You are correct, these merit some further mention in our perspective. As you and I have discussed (e.g., at WLF6 in Florence), data-driven models are a great approach for replicating some observed relationships. Indeed, with enough data they are providing further insights and tools to quantify the spatial and temporal occurrence of landslides in well studied areas (e.g., Lombardo et al., *Earth Sci. Rev.*, 2020). While we may have underemphasized these approaches in our perspective, we also wanted to avoid a detailed discussion or voicing any criticism of methods we have not fully explored in our own research; the paper is a perspective, not a critique (or even a comment piece). However, as is perhaps more appropriate for this critical discussion, we take the opportunity here to point out that data-driven models can be great where there is data, but naïve machine learning models are strongly biased by the data they consider and thus may struggle to capture the observed hydrologic response and landslide triggering in areas *without* sufficient data. This data-bias effect may be less of an issue with physically based models, that capture the (incompletely) known physics of infiltration and drainage (albeit there are other biases we discuss related to the assumption of diffuse flow equations). Regarding the specific papers you mention, we have examined these and others to inform our responses to each set of papers below.

Yes, Stefan Steger's works (i.e., *NHESS,* 2023; *Front. Geosci*, 2024) are indeed very relevant, so are other space-time modeling studies authored by you and your colleagues (e.g., Lombardo et al., *Earth Sci. Rev.,* 2020) as well as contrasting efforts (e.g., Bordoni et al., *Landslides,* 2019). We had neglected to draw the connection between landslide forecasting used in early warning and promising potential shown by data-driven space-time modeling approaches such as these. As you all have noted, there is an issue of data bias and Stefan has proposed novel ways to deal with this (i.e., Steger et al., *Geosci. Front.*, 2024). However, some issues remain, including the need for extensive data on landslide timing, as well as the assumption that future triggering conditions will be controlled in a similar manner to past ones. In some cases these can be overcome, in others, more data is needed. We can revise the manuscript to reflect our perspective on the potential value of these novel contributions and acknowledge that our paper focuses on how to inject hydrologic understanding from in-situ monitoring and simplified process-based modeling studies.

You also note the excellent body of work by Stanley, Kirschbaum, and others, which we must also point out does not provide an explicit pathway for early warning of landslides, but rather for "nowcasting" of global landslides conditions. Still, they initially used the AR7 weighted rainfall index for LHASA V1 (Kirschbaum and Stanley, *Earths Future*, 2018) and then transitioned to a probabilistic approach using the 99% percentile of extreme daily rainfall for V2 (Stanley et al., *Front. Earth Sci*., 2021). Furthermore, in V2 they now consider SMAP data for antecedent conditions, but as we discuss, the satellite estimates of hillslope hydrology are often inappropriate for capturing hillslope drainage conditions (Thomas et al., *WRR,* 2019), not to mention the considerable latency of ~3 days. So, overall, theirs is great uniform approximation for generalizing the role of past rainfall in landslide triggering globally, but in our view this broad approach precludes practical application for local or regional landslide early warning that reflects an improved hydrologic understanding. While the global model output is interesting in that it can be applied everywhere to facilitate uniform risk and exposure assessments (e.g., Emberson et al., NHESS, 2020), it is potentially quite inappropriate for informing mitigation measures at local scales. For example, Marc et al. (*Earth Interactions*, 2022) nicely demonstrated that even after extensive testing a simple rainfall recurrence-interval based threshold has great promise for only about half of the events they considered. We suggest that for actually improving local or regional scale warnings that lead to actionable information, quantifying the local and regional factors that control antecedent conditions, infiltration, and drainage dynamics, is a promising path ahead. In the revised manuscript we will strive to stress these nuances for landslide warning without delving into the advantages or limitations of the LHASA model.

Thank you for pointing us and other readers to the study of Pudasaini and Krautblatter (2021), which is fascinating and indeed injects an unprecedented degree of mathematical formalism into landslide runout behavior. However, for this particular study we don't see a clear connection with our section on extrapolating through space and time, or how this relates to landslide warning, or even how it is relevant to the problem of *when* or *where* landslides will initiate. Perhaps you can more eloquently make this connection between improved runout modeling and landslide warnings in a future publication, but for the present manuscript we are not likely to include this in a meaningful way to add to our discussion about hydrologic information.

Overall, for the reasons discussed above regarding the current state of space-time modeling (e.g., Steger et al., 2024) and global-scale now-casting (i.e., Stanley et al., 2021), we advocate for a wider global network of hillslope hydrological monitoring, precisely so that there are more grounds for understanding and testing landslide hydro-climatology and how those may change in the future under climate change scenarios. Perhaps data-driven models are ultimately a good way to leverage that data more effectively than we have with more site-specific physically based or empirical thresholds models, but that is an argument for a later date when more of such data are available. For the present paper, I intend to read, and in some cases re-read the articles you suggest, and my coauthors and I will consider how to integrate further discussion on the advantages and drawbacks of these data-driven approaches.

Once again, we are very grateful for your time and keen insights, and certainly think an in-depth exploration of your own perspective would be equally worthwhile for some future publication. Personally, I am eager for further conversations with you on these topics moving forward.

Warmest regards,

Ben, Thom, Roberto, and Manfred

---

## Author Comment (AC2)

Dear Wei-Li,

Thank you very much for your positive assessment and some very constructive suggestions for improvement. We are glad that you agree about the importance of hydrologic information and given your expertise with hydrological field instrumentation and mathematical modeling, we are grateful for your informed input.

Regarding your first comment on Lines 298-232 (relative wetness): yes, you are correct. Theoretically each individual hillslope may have a critical pore-water pressure (or saturated thickness) that will trigger failure at that specific location. However, it is quite challenging to identify a critical value for a given state variable based on monitoring at only one location, and then that value may not apply effectively for characterizing failure potential throughout the rest of the study area. Indeed, across a heterogeneous landscape there are a multitude of hydrologic responses, not to mention spatially variable rainfall that are a factor. While there is a lot to learn from precise observations at a site that undergoes landsliding (e.g., Montogmery et al., *WRR*, 2009; Godt et al., *GRL*, 2009; Mirus et al., *WRR*, 2017; Liang, *J. Hydrol.*, 2020), there is also some value in long-term monitoring that can survive one or multiple landslide events. In fact, as Wicki et al. (*NHESS*, 2023) and others have demonstrated, it's possible to leverage hydrologic information from non-landslide prone areas to improve hydrometeorological thresholds for landslide triggering. Thus, what we have found through our combined research is that for warning purposes it is more important to understand the *relative change* in wetness throughout a landscape, rather than isolating a critical threshold value of a given state variable. Additionally, precise calibration of sensors to measure accurate volumetric water contents are important for some modeling calculations, but may be of limited value for warnings compared to capturing the range of responses to multiple storms (see Wicki et al., *Landslides*, 2020). As a consequence, we suggest that for LEWS, identifying a monitoring location and selecting measured variables that can best capture the full range of those hillslope wetness dynamics, as this may provide the most informative guide for generalized landslide forecasting. This is one of the main reasons for including our Sitka data example (Figure 1), since the soil moisture data in Soil Pit #2 exhibit very little dynamic responses and are thus of limited value, whereas the piezometer in SP #2 greatly enhances comprehension of the hillslope storage and drainage dynamics. Thanks for your observation, we will revise this section to make our argument clearer and more transparent.

Regarding your suggestion for reformatting Figure 1: OK, we agree that is doable and worthwhile. Probably it makes sense to include three panels with (a) rainfall and temperature, (b) volumetric water content at SP1 and SP1, and (c) pore-water pressures at SP1 and SP2. These revisions will allow readers to directly compare the responses and information between the two soil pits with appropriate y-axis scales. Thanks for this suggestion!

Regarding your observation on information in Fig. 1 and Lines 139-145: yes, the contrast you describe between pore-water pressures and soil saturations in SP1 vs. SP2 is indeed consistent with our intended message. In fact, the ephemeral saturation in SP1, an area that is usually *unsaturated*, has the strongest correlation as an indicator of when landslides are likely (see graphs below), but because of the ephemeral nature cannot provide much lead-time in advance of potential landsliding. We can certainly revise the text to better emphasize the contrasting utility of the two soil pits and types of monitoring equipment at each location.

[Figure]

**Discussion Figure 1.** Cumulative probability distribution and landslide occurrences near Sitka, Alaska (AK), for different observations of (a) shallow groundwater pressures measured every five minutes by piezometers in SP1 and SP2, (b) volumetric soil-water content measured every five minutes by four probes in SP1 and SP2, and (c) daily maximum accumulation of rainfall during any three-hour period. Note that all landslide events listed from September 2019 - November 2020 occurred within 2 km of Sitka, AK, (Patton et al., *NHESS*, 2023). Note that the November 2023 event resulted in widespread landsliding over 150km to the southeast across Prince of Wales Island and in Wrangle, AK, whereas the December 2020 event occurred 250 km to the northeast in Haines, AK (Darrow et al., *Landslides*, 2022).

Regarding your comment on Figure 2 (contrasting hydrometeorological thresholds): yes, your additional description is helpful. The inference we wanted to make is that despite previous subjective modeling choices, one might generally expect a threshold like the one shown in red, where an increasing antecedent saturation to result in a monotonically decreasing rainfall amount required to exceed the threshold (or for increasing rainfall a corresponding decrease in the required antecedent saturation). Perhaps it was not clear enough that this "general threshold" is merely a suggestion based on this concept, or a hypothesis we propose for further testing with further data and modeling. As you note, additional exploration of contrasting regions, hydroclimatologies, and other factors should help refine what threshold format(s) are more (and less) appropriate in different settings. We can further revise this section to clarify those points.

Once again, thank you for the suggestions, these will help further improve the clarity of our manuscript and emphasize our main points related to the value of hillslope hydrologic information.

Warmest regards,

Ben, Thom, Roberto, and Manfred

---

## Author Response (AR1)

Dear Professor Tarolli,

Thank you very much for the opportunity to submit this invited commentary to NHESS and for facilitating the engaging discussion surrounding our initial submission. As we outlined in our response to the two reviewers, Professors Luigi Lombardo and Wei-Li Liang, we appreciate their feedback and suggestions and have made the recommended changes summarized below.

In response to concerns from Reviewer #1, Luigi Lombardo:

1) Substantially reduced the self-citations of works by Ben Mirus, from _seven first-authored publications down to only three_, and from _15 co-authored publications down to nine_. The remaining citations are critical to our main arguments and narrative in this perspective piece. Moreover, we have made a concerted effort to _supplement these with additional citations_ whenever appropriate.

2) Added some _discussion of emerging approaches in space-time landslide modeling_ (i.e., Steger et al., 2024) and another new process-based approach (i.e., Perkins et al., NHESS) to both the "extrapolation in space" and "towards improved forecasting" sections of the paper.

In response to suggestions from Reviewer #2, Wei-Li Liang:

1) _Revised Figure 1_ to include different panels to distinguish between VWC and PWP. The resulting figure is much easier to interpret.

2) Added a _new figure to demonstrate the role of relative wetness_ and comparing the value of information of different state variables (now Figure 2).

3) Added text to _clarify that the "generalized threshold" in earlier version of Figure 2 (now Figure 3) is an untested hypothesis_ and that further studies will either reveal a range of suitable formats for hydrometeorological thresholds or potentially demonstrate that approaches need to be tailored to individual settings.

We are convinced that these relatively modest changes have greatly improved the manuscript and we hope you will find it suitably revised for publication in NHESS. Once again thanks to you and the reviewers for your time and input throughout the review and discussion process. We and look forward to your decision.

Warmest regards,

Ben Mirus, Thom Bogaard, Roberto Greco, and Manfred Stähli